# Long-Term Mortality after *Histoplasma* Infection in People with HIV

**DOI:** 10.3390/jof7050369

**Published:** 2021-05-08

**Authors:** Joseph Cherabie, Patrick Mazi, Adriana M. Rauseo, Chapelle Ayres, Lindsey Larson, Sasinuch Rutjanawech, Jane O’Halloran, Rachel Presti, William G. Powderly, Andrej Spec

**Affiliations:** Division of Infectious Diseases, Department of Medicine, Washington University School of Medicine, St. Louis, MO 63110, USA; jcherabie@wustl.edu (J.C.); pmazi@wustl.edu (P.M.); a.rauseoacevedo@wustl.edu (A.M.R.); ayreschapelle@wustl.edu (C.A.); lindseylarson@wustl.edu (L.L.); sasinuchr@wustl.edu (S.R.); janeaohalloran@wustl.edu (J.O.); prestir@wustl.edu (R.P.); wpowderly@wustl.edu (W.G.P.)

**Keywords:** HIV, *Histoplasma*, antiretroviral therapy, mortality

## Abstract

Histoplasmosis is a common opportunistic infection in people with HIV (PWH); however, no study has looked at factors associated with the long-term mortality of histoplasmosis in PWH. We conducted a single-center retrospective study on the long-term mortality of PWH diagnosed with histoplasmosis between 2002 and 2017. Patients were categorized into three groups based on length of survival after diagnosis: early mortality (death < 90 days), late mortality (death ≥ 90 days), and long-term survivors. Patients diagnosed during or after 2008 were considered part of the modern antiretroviral therapy (ART) era. Insurance type (private vs. public) was a surrogate indicator of socioeconomic status. Out of 54 PWH infected with histoplasmosis, overall mortality was 37%; 14.8% early mortality and 22.2% late mortality. There was no statistically significant difference in survival based on the availability of modern ART (*p* = 0.60). Insurance status reached statistical significance with 38% of survivors having private insurance versus only 8% having private insurance in the late mortality group (*p* = 0.05). High mortality persists despite the advent of modern ART, implicating a contribution from social determinants of health, such as private insurance. Larger studies are needed to elucidate the role of these factors in the mortality of PWH.

## 1. Introduction

Histoplasmosis is the most prevalent endemic mycosis in the United States; an estimated 50,778 histoplasmosis-associated hospitalizations occurred between 2001 and 2012 with a trend towards increased incidence [1]. The incidence of histoplasmosis among adults in the US is highest in the Midwest with 5.1 cases per 100,000 population and common worldwide among people with HIV/AIDS (PWH) without access to anti-retroviral therapy (ART) [2].

Histoplasmosis carries a high 90-day mortality of 16%, with mortality increasing among immunocompromised individuals up to 24% [3]. Previous studies have found that histoplasmosis, specifically disseminated histoplasmosis, carries a high mortality among PWH greater than 30%, although many of these studies were performed prior to the era of modern ART [4,5]. In other opportunistic mycoses, late mortality appears to be significantly higher than the early mortality initially observed [6].

While multiple studies have assessed factors associated with early mortality from histoplasmosis among PWH [7,8,9,10], to the best of our knowledge no studies have thus far looked at factors associated with long-term mortality, particularly within the era of modern ART.

## 2. Materials and Methods

This is a retrospective cohort study of patients diagnosed with *Histoplasma* infection at Barnes-Jewish Hospital (BJH) in St. Louis, Missouri. BJH is a 1368-bed tertiary care academic hospital. While located in an urban setting, BJH has a large suburban and rural catchment area. This study was approved by the Washington University School of Medicine Human Research Protection Office, with a waiver of informed consent.

The cohort consisted of all individuals diagnosed with *Histoplasma capsulatum* infection aged 18 or older within BJH from 1 January 2002 until 31 December 2017. *Histoplasma* infection required one of the following: (1) positive *Histoplasma* antigen from urine, serum, CSF, or other body fluid; (2) microbiologic isolation of *H. capsulatum* from any source; (3) coding for International Classification of Diseases (ICD) 9th (115.x) or 10th (B39.X) codes for *Histoplasma* infection; or (4) positive *Histoplasma* antibody. Cases were confirmed independently by two study investigators to ensure cases met criteria for proven or probable histoplasmosis by MSG/EORTC criteria [11]. Exclusion criteria consisted of the following: (1) if antigen or antibody testing was positive in the setting of another fungal infection confirmed by culture or histopathology; (2) if an antigen was positive at a low or indeterminate level and the case was determined to be a false positive by the treating infectious diseases team at the time and/or by the two study investigators during independent review; (3) if there was a diagnosis of presumed ocular histoplasmosis syndrome without other evidence of *Histoplasma* infection; and (4) if the patient presented with fibrosing mediastinitis. PWH were then sub-selected if they had a positive 3rd or 4th generation HIV screening test with positive confirmatory HIV viral load test. Additional information pertaining to each patient’s HIV infection was obtained from the patients’ charts, including CD4 count and HIV viral load at time of *Histoplasma* infection diagnosis, HIV viral load at the time of last encounter, anti-retroviral experience prior to diagnosis, antiretroviral prescribed at the time of diagnosis, insurance status, history of other opportunistic infections (OI), history of substance use, and history of psychiatric illness. Data were collected through automated extraction from the electronic medical record and manual chart review. Other participant data collected included patient demographics, clinical presentation, organ system involvement in infection, and mortality.

The primary outcome for our study was all-cause late mortality, defined as deaths occurring at or after 90 days of histoplasmosis diagnosis. The secondary outcome was all-cause early mortality, defined as deaths occurring within 90 days of histoplasmosis diagnosis. Date of death was obtained from the electronic medical record and Social Security Death Index. The Social Security Death Index includes deaths through 2014 with deaths after 2014 obtained through the electronic medical record system. If there was no date of death on record, then it was assumed the patient survived until the last day the patient interacted with the healthcare system and was right-censored after that date. Pre-modern ART was defined as a participant having been diagnosed with histoplasmosis before 2008, and modern ART was a participant having been diagnosed from 2008 onwards, with 2008 marking the advent of darunavir and integrase inhibitors as standard ART. A participant was considered virally suppressed if they had a HIV viral load < 50 copies/mL. Insurance status was categorized into two groups: those with private insurance and individuals with government insurance (Medicare or Medicaid) or no insurance.

Categorical variables were analyzed using Pearson Chi squared analysis. Continuous variables were analyzed using Mann–Whitney U non-parametric testing to account for small numbers and lack of normal distribution. *p*-values ≤ 0.05 were considered statistically significant. All statistical analysis was performed using SPSS V25 (IBM, Armonk, NY, USA).

## 3. Results

Fifty-four individuals were found to have both histoplasmosis as well as HIV infection within our study period; 68.5% were male and 59.3% were Black. There was no statistically significant difference between survival groups with respect to age (*p* = 0.28), gender (*p* = 0.91) or race (*p* = 0.11) (Table 1).

Overall mortality was 37%. A total of 14.8% (*n* = 8) of individuals expired within 90 days of histoplasmosis diagnosis with a median length of survival of 13.5 days (IQR 2.5–41), while 22.2% (*n* = 12) expired on or after 90 days with a median length of survival of 302 days (IQR 153–710.5). Death occurred within 30 days in 9.3% of individuals (*n* = 5), in 31–90 days in 5.5% of individuals (*n* = 3), in 91–365 days in 13% of individuals (*n* = 7), and greater than 365 days in 9.3% of individuals (*n* = 5). Death occurring within 1 year had a median survival of 86 days (IQR 6–177) and after 1 year had a median survival of 798 days (IQR 418–993). 

Private insurance was a factor more common to the survival group at 38% compared to 8% of the late mortality group that had private insurance (*p* = 0.05) (Table 2). There was no significant difference between survivors and late mortality individuals with respect to history of substance use (survivors *n* = 13, 38% survivors vs. late mortality individuals *n* = 2, 17%, *p* = 0.17) or history of psychiatric illness (survivors *n* = 4, 12%, vs. late mortality individuals *n* = 0, 0%, *p* = 0.21).

### 3.1. Disease Presentation

Disseminated histoplasmosis was the most common disease presentation within all groups, occurring in 76% of survivors (*n* = 26), 63% of individuals with early mortality (*n* = 5), and 92% of individuals with late mortality (*n* = 11), with no statistically significant difference between the three groups (*p* = 0.29) (Table 1). Localized pulmonary infection was common within all three groups (*n* = 37, 68.5%, *p* = 0.67), and central nervous system (CNS) infection was seen in three patients, all of whom survived. As for presenting symptoms, fever was the most common (74.1%), followed by gastrointestinal symptoms (64.8%) and cough (48.1%).

### 3.2. HIV Characteristics

While the median time in years from HIV diagnosis to histoplasmosis diagnosis in survivors was 3.3 years shorter than among late mortality individuals (0.2 years, IQR 0–13.6 years vs. 3.5 years, IQR 0.1–5.3 years), this difference was not statistically significant (*p* = 0.87) (Table 2). Having a new HIV diagnosis at the time of *Histoplasma* diagnosis was more common among survivors (*n* = 18, 53%) compared to late-mortality individuals (*n* = 3, 25%), but this also did not meet statistical significance (*p* = 0.10).

HIV viral load at the time of diagnosis with histoplasmosis did not vary significantly between survivors (5.5 median log copies/mL, IQR 4.4–6.2) and individuals with late mortality (5.2 median log copies/mL, IQR 3.8–5.5) (*p* = 0.25). Similarly, there was no statistically significant difference in viral load suppression (survivors *n* = 1, 3% vs. late mortality *n* = 2, 17%) (*p* = 0.10) or median CD4 count at the time of *Histoplasma* infection diagnosis (median CD4 12 cells/mm^3^, IQR 6–46 in survivors vs. median CD4 26 cells/mm^3^, IQR 8–52 in late mortality individuals, *p* = 0.52).

Median HIV viral load at last observation was significantly lower in survivors (2 log copies/mL, IQR 0–4.5) compared to the late mortality individuals (4.1 log copies/mL, IQR 2.6–5.5) (*p* = 0.01). Survivors (*n* = 14, 41%) were more than twice as likely to have HIV viral load suppression at the time of last observation compared to individuals with late mortality (*n* = 2, 17%); however, this difference was not statistically significant (*p* = 0.13). 

### 3.3. ART Experience and Opportunistic Infections

ART experience made no difference with respect to survival (*n* = 14, 41% of survival group vs. *n* = 7, 58% of late mortality group, *p* = 0.19), nor did being diagnosed with *Histoplasma* infection in the era of modern ART (*n* = 12, 35%, of survival group vs. *n* = 4, 33% of late mortality group, *p* = 0.90). Previous opportunistic infections (OIs) occurred in 50% of the survivor group (*n* = 17) vs. 75% of the late mortality group (*n* = 9), with no significant difference in OI occurrence between the two groups (*p* = 0.13). The most common other opportunistic infections overall were *Pneumocystis* pneumonia, followed by oral candidiasis and *Cytomegalovirus* infection. Of all the individuals with late mortality and OIs, three had no death certificates to indicate cause of death, while three had one or more OIs other than histoplasmosis active at the time of death.

## 4. Discussion

This is the first study assessing the long-term mortality of histoplasmosis in a US population of PWH. Despite the resource-rich environment of US healthcare and most of our study occurring during the modern era of ART, overall mortality was 37%. This is a similar mortality rate to the resource-limited environments of French Guiana, Guatemala, and Brazil—all-cause mortality of 41%, 43.6%, and 30.2%, respectively [11,12,13]. In the French Guiana and Guatemalan cohorts, approximately 75% of deaths occurred within a year of histoplasmosis diagnosis. Our study showed similar findings to these studies, suggesting that the first 12 months is a critical period for PWH. 

The specific cause of each patient’s death is uncertain due to many OIs being listed as active at the time of death with causes of death being multifactorial. This is often the case in patients with uncontrolled HIV. Despite this, we identified that HIV viral load at the time of diagnosis with histoplasmosis did not vary significantly between survivors and individuals with late mortality. Similarly, there was no statistically significant difference in viral load suppression or median CD4 count at the time of *Histoplasma* infection diagnosis. Patients presenting in the French Guiana and Guatemala cohorts also had uncontrolled HIV with median CD4 counts of 31 and 25, respectively. 

The advent of integrase inhibitors in 2007–2008 ushered in the modern era of ART. Modern ART is better tolerated, leads to more rapid viral suppression, and has higher uptake than historical formulations [14,15]. While the advent of modern ART has had an effect on the mortality of PWH with other mycoses such as *Cryptococcus* [6], in our cohort, being diagnosed with histoplasmosis in the era of modern ART had no effect on survival.

To our knowledge, our study is the first to look at social factors such as private insurance, mental health, and substance use on the mortality of PWH with histoplasmosis. A statistically significant improvement in mortality was observed in patients having private insurance vs. government or no insurance. Health insurance itself is a reflection on socio-economic status, especially within the US, where individuals with insurance report significantly greater access and quality of health services compared to individuals who are uninsured, especially with respect to accessing specialist appointments when private insurance was compared to Medicaid and uninsured individuals [16]. Negative outcomes are common among PWH who are uninsured, especially with regard to limited access to indicated ART regimens, case management, and mental health services [17,18].

Delayed mortality from OIs in PWH is not unique to just histoplasmosis. In PWH diagnosed with *Cryptococcus* infection, 66% of deaths occurred after 90 days [6]. Similar to our study, survivors were more likely to have private insurance. In contrast to our study, the authors found improved mortality associated with viral suppression at last follow up. Additionally, they found an improvement in outcomes in the modern ART era and higher rates of substance use and mental illness in non-survivors [6]. The opposite was seen in our study, with survivors having higher rates of substance use and a history of psychiatric illness, and although neither finding was statistically significant, these findings suggest these factors may not affect mortality in histoplasmosis patients in the same way as cryptococcosis.

Other OIs in the era of modern ART do not show similar delayed mortality as observed in histoplasmosis and cryptococcosis. A study of 1264 Taiwanese PWH in the era of modern ART found 21% of patients had OIs, with the most common being *P. jirovecii* (43.4%), followed by *Cytomegalovirus* (10.4%), tuberculosis (8.1%), and candidiasis (6.9%). The majority of OI events (91.7%) developed within 90 days and all-cause mortality was 4.4%; 83.9% being OI related etiologies with no significant difference between mortality rate at 90 days, between 91 and 180 days, and >180 days [19]. These data contrast with our cohort of PWH with histoplasmosis as their diagnosed OI, where the majority of mortality occurred more than 90 days after diagnosis.

Small sample size limited our ability to perform multivariate analyses among all three groups. The early mortality group was too small, with wide confidence intervals in any comparison, limiting comparisons to the survivor and the late mortality groups. Another limitation is that our data did not capture whether a patient’s insurance status changed after their hospitalization. Insurance status is subject to change over time as the result of many factors; however, due to the size of our study, we considered this a definitive variable. This study was performed at a single, academic medical center located in the Midwest, which may limit the generalizability of results to other centers with variable endemicity of histoplasmosis and to regions where different OIs are more endemic. Lastly, and possibly most importantly, we were limited by our lack of definitive cause of death due to the retrospective nature of the study.

In conclusion, the mortality of PWH diagnosed with histoplasmosis remains high, even in the resource-rich environment of the US medical system, with comparable gross mortality to similar cohorts in more resource-limited settings [12,13,20]. Despite access to accurate diagnostics, modern ART, and effective antifungal treatment regimens, our US cohort did not trend towards improved mortality as we transitioned in the era of modern ART, unlike the cohorts from French Guiana, Guatemala, and Brazil. Patients with government or no insurance were five times more likely to have a late death compared to patients with private insurance. Though these are complex medical situations with many confounders, we hypothesize private insurance is a surrogate for a patient’s ability to access and utilize the US medical system. That PWH in our cohort became immunocompromised enough to be diagnosed with histoplasmosis and then to experience delayed mortality beyond the duration of histoplasmosis treatment is likely a failure of the US medical system to maximize the care this population needs. We recommend additional research including broadening our cohort to other resource-rich settings to further test our hypothesis and more fully elucidate the deficiencies in our HIV care model.

## Figures and Tables

**Table 1 jof-07-00369-t001:** Baseline characteristics and presentation of PWH with histoplasmosis (*n* = 54) among three groups (survivors, early mortality, and late mortality) 2002–2017.

	Survived *n* = 34 (63.0%)	Early Mortality *n* = 8 (14.8%)	Late Mortality *n* = 12 (22.2%)	*p*-Value
Male	23 (67%)	6 (75%)	8 (67%)	0.91
Age (years, median, IQR)	43 (32, 51)	41 (36, 49)	36 (25, 40)	0.28
Race				0.11
Black	16 (47%)	7 (87%)	9 (75%)	
Non-Black	17 (50%)	1 (13%)	2 (17%)	
Site of infection				
CNS	3 (9%)	0 (0%)	0 (0%)	0.39
Pulmonary	24 (71%)	6 (75%)	7 (58%)	0.67
Bloodstream	15 (44%)	3 (38%)	8 (67%)	0.33
Disseminated disease	26 (76%)	5 (63%)	11 (92%)	0.29
Presenting symptoms				
Fever	26 (76%)	4 (50%)	10 (83%)	0.22
Cough	17 (50%)	4 (50%)	5 (42%)	0.88
Night sweats	10 (29%)	1 (13%)	5 (42%)	0.38
Dyspnea	13 (38%)	3 (38%)	6 (50%)	0.76
Chest pain	6 (18%)	0 (0%)	3 (25%)	0.33
Arthralgias	3 (9%)	0 (0%)	1 (8%)	0.69
Dysphagia	5 (15%)	2 (25%)	2 (17%)	0.78
Weight loss	22 (65%)	5 (63%)	5 (42%)	0.37
GI symptoms	23 (68%)	4 (50%)	8 (67%)	0.64

**Table 2 jof-07-00369-t002:** Baseline HIV-related characteristics of PWH with histoplasmosis (*n* = 46) among two groups (survivors and late mortality) 2002–2017.

	Survived (*n* = 34)	Late Mortality (*n* = 12)	*p*-Value
Median CD4 count (cells/mm^3^, median, IQR)	12 (6, 46)	26 (8, 52)	0.52
Median HIV viral load at diagnosis (copies/mL, median, IQR)	5.5 (4.4, 6.2)	5.2 (3.8, 5.5)	0.25
Median HIV viral load at last observation (copies/mL, median, IQR)	2.0 (0, 4.5)	4.1 (2.6, 5.5)	0.01
New HIV diagnosis at diagnosis	18 (53%)	3 (25%)	0.10
Median time between HIV and *Histoplasma* diagnoses (years, median, IQR)	0.2 (0, 13.6)	3.5 (0.1, 5.3)	0.87
ART-experienced	14 (41%)	7 (58%)	0.19
Modern ART	12 (35%)	4 (33%)	0.90
HIV viral load suppressed at last observation	14 (41%)	2 (17%)	0.13
HIV viral load suppressed at diagnosis	1 (3%)	2 (17%)	0.10
History of previous OIs	17 (50%)	9 (75%)	0.13
*Pneumocystis* pneumonia	10 (29%)	2 (17%)	
Oral Candidiasis	4 (12%)	2 (17%)	
CMV	2 (6%)	2 (17%)	
Private insurance	13 (38%)	1 (8%)	0.05
History of substance use	13 (38%)	2 (17%)	0.17
History of psychiatric illness	4 (12%)	0 (0%)	0.21

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
