# Peer review of "Long-Term Mortality after Histoplasma Infection in People with HIV"

_jof, 2021, doi:10.3390/jof7050369_

Round 1
Reviewer 1 Report
General comments
This study includes an analysis of long term mortality in HIV patients with histoplasmosis in a small center in St. Louis, Missouri. Although the authors consider multiple factors and generate several statistical analyses, no real conclusion can be made with the data obtained. They observed that HIV viral load quantification at time of histoplasmosis diagnosis or antiretroviral experience was not different between survival and late mortality groups, but they suggest that late mortality is more likely to occur in patients with no insurance versus those with private insurance, which is not surprising. The authors do not seem to find a strong conclusion or factor related with the mortality of the patients included in this study, mainly because the number of patients is so small that no real comparisons can be made, and also because the cause of death is unknown. In addition, there is no data whatsoever about the patients being treated or not with antifungal agents when histoplasmosis was diagnosed, or if there was antibiotic or antifungal treatment for other opportunistic infections, which seemed very common in the studied cohort.
Why is late mortality data in patients with HIV and diagnosed with histoplasmosis important if the cause of death is not determined? Is histoplasmosis monitored?
Why the authors split the races only in Black and non-Black? Were there patients of other ethnicities?
Also, replace “mm3” by “mm3” throughout the text.
Specific comments
Line 14: Introduce “ART”
Line 49: How were the isolates identified to the species level?
Lines 127-129: this sentence is confusing. What do the authors mean by “new HIV diagnoses”?
Lines 137-141: Does this suggest that the patients died from HIV? Was histoplasmosis monitored?
Line 146: introduce “OI”.
Line 214: no antifungal treatment has been mentioned in the text.
Author Response
Point 1: There is no data whatsoever about the patients being treated or not with antifungal agents when histoplasmosis was diagnosed, or if there was antibiotic or antifungal treatment for other opportunistic infections.
Response 1: Thank you for this comment. We felt that adding information on antifungal treatment deviated from the main topic, observing late mortality in patients co-infected with HIV and histoplasmosis. As a large tertiary care hospital, all patients were treated for histoplasmosis upon diagnosis according to standard of care.
Point 2: Why is late mortality data in patients with HIV and diagnosed with histoplasmosis important if the cause of death is not determined? Is histoplasmosis monitored?
Response 2: We used all-cause mortality as an outcome as these cases often have multiple comorbidities and it is not very simple to say that one thing was the cause of death as opposed to the other. We performed a chart review to see if other OIs were implicated and those results are listed in lines 151 - 156. Further explanations are given in line 166-168.
Point 3: Why did the authors split the races only in Black and non-Black? Were there patients of other ethnicities?
Response 3: There were patients of other ethnicities in our cohort but once that was broken down further, it wouldn't allow for further analysis as many of the ethnicity groups included only one individual. Additionally, dividing race by Black and non-Black is a reflection of how Black individuals are disproportionately affected by HIV.
Point 4: Minor edits for grammar correction and sentence structure: replace “mm3” by “mm3” throughout the text, introduce “ART”, introduce “OI”
Response 4: Thank you for bringing this to our attention. We have made the appropriate changes for mm3, ART, and OIs.
Point 5: How were the isolates identified to the species level?
Response 5: Histoplasmosis in the United States is caused by Histoplasma capsulatum, our identification strategies are listed in our methods section.
Point 6: What do the authors mean by “new HIV diagnoses?” This sentence is confusing.
Response 6: Thank you for this comment. We have revised this sentence for better clarity.
Point 7: Does [the following sentence] suggest that the patients died from HIV? Was histoplasmosis monitored?
“Median HIV viral load at last observation was significantly lower in survivors (2 log copies/ml, IQR 0-4.5) compared to … however this difference was not statistically significant (p=0.13)”
Response 7: Thank you for this comment. Patients in the late mortality group likely had uncontrolled HIV compared to survivors at their last observation, but we can’t attribute death to HIV alone.
Point 8: No antifungal treatment has been mentioned in the text.
Response 8: Please refer to Response 1.
Reviewer 2 Report
This is a retrospective study where, factors associated to long-term mortality (after 90 days) in 54 HIV individuals after histoplasmosis were analysed in a single US tertiary-care centre. These are some recommendations for authors:
- Table 1 and 2, please clarify units for continuous variables example, years for age and logs for viral load?
- Authors need to discuss more about the differences between public and private insurance, assuming all private insurance equals to better attention is misleading and doesn't represent reality of other places apart from the BJH. Details on where the main differences are, should be discussed and/or if info available be included within the analysis (example: earlier diagnosis, more/better diagnostic tools, better access to ID consultants, follow up/lack of follow up, time from symptoms to health care first visit, among others)
- I'd recommend authors these references PMID: 30205586 DOI: 10.3390/jof4030109, PMID: 30668766 DOI: 10.1093/mmy/myy143 for more comparative data on HIV and histoplasmosis experience in Latam.
- Even when authors didn't see significance, there is a trend for higher survival rate when histoplasmosis was diagnosed along HIV. Could this mean that histoplasmosis is being diagnosed as part of OI screening in new HIV diagnosed individuals? and then not suspected if not in that context?
- Was IRIS diagnosed among patients in the study? is it possible to know if impacted long-term mortality?
Author Response
Point 1: Table 1 and 2, please clarify units for continuous variables example, years for age and logs for viral load?
Response 1: Thank you we have made the appropriate changes.
Point 2: Authors need to discuss more about the differences between public and private insurance, assuming all private insurance equals to better attention is misleading and doesn't represent reality of other places apart from the BJH. Details on where the main differences are, should be discussed and/or if info available be included within the analysis (example: earlier diagnosis, more/better diagnostic tools, better access to ID consultants, follow up/lack of follow up, time from symptoms to health care first visit, among others).
Response 2: Thank you for this comment. We have expanded this section as well as included a new citation for better clarity.
Point 3: I'd recommend authors these references PMID: 30205586 DOI: 10.3390/jof4030109, PMID: 30668766 DOI: 10.1093/mmy/myy143 for more comparative data on HIV and histoplasmosis experience in Latam.
Response 3: Thank you for this comment. We have included data on HIV and histoplasmosis in a Brazilian cohort after reviewing these references.
Point 4: Even when authors didn't see significance, there is a trend for higher survival rate when histoplasmosis was diagnosed along HIV. Could this mean that histoplasmosis is being diagnosed as part of OI screening in new HIV diagnosed individuals? and then not suspected if not in that context?
Response 4: This is a possibility indeed and there is a trend, however, we chose not to focus on this to give more room for analysis on significant findings.
Point 5: Was IRIS diagnosed among patients in the study? is it possible to know if impacted long-term mortality?
Response 5: IRIS was not directly looked into within this study, with new HIV diagnoses occurring in less than half of the population, and often times it would be correlated with early mortality rather than late mortality which was not the focus of this study.